# Role of the PhoP/PhoQ Two-Component Regulatory System in Biofilm Formation in Acid-Adapted *Salmonella typhimurium*

**DOI:** 10.3390/foods14244344

**Published:** 2025-12-17

**Authors:** Huixuan Yang, Xueqing Jiang, George-John E. Nychas, Kehui Yang, Pengcheng Dong, Yimin Zhang, Lixian Zhu, Yunge Liu

**Affiliations:** 1Lab of Beef Processing and Quality Control, College of Food Science and Engineering, Shandong Agricultural University, Tai’an 271018, China; 2International Joint Laboratory of Shandong Province for Quality and Safety Control of Livestock and Poultry Products and Intelligent Manufacturing, Tai’an 271018, China; 3Department of Food Science and Human Nutrition, Agricultural University of Athens, Iera Odos 75, 11855 Athens, Greece

**Keywords:** *Salmonella typhimurium*, PhoP, two-component regulatory system, biofilm, acid stress

## Abstract

*Salmonella typhimurium* is a global foodborne pathogen, and controlling its persistence is critical for public health. This study investigated the regulatory role of the PhoP/PhoQ two-component system (TCS) in biofilm formation under the acid adaptation condition. A *phoP* deletion strain (Δ*phoP*) was constructed and compared with the wild type (WT) after acid induction (pH 5.4). Without acid adaptation, Δ*phoP* and WT showed similar acid tolerance and biofilm formation. However, after acid induction, Δ*phoP* exhibited markedly reduced biofilm formation, swimming ability, metabolic activity, and extracellular polymer production. RNA-seq analysis further revealed defects in Δ*phoP* under acid-induced conditions: (i) first leads to downregulation of lipopolysaccharide biosynthesis, peptidoglycan synthesis, and cationic antimicrobial peptide resistance pathways, thereby weakening the bacteria’s envelope modification capacity and structural stability; (ii) it also disrupts signal regulations in acidic environments, further impairing energy metabolism, flagellar function, and chemotaxis, thereby affecting bacterial adhesion capacity and environmental adaptability. These results demonstrate that under acid adaptation, the PhoP/PhoQ TCS is critical for coordinating cell envelope remodelling, energy metabolism, and motility to support biofilm formation in *S*. *typhimurium*. Understanding the contribution of this system to biofilm formation is essential for addressing the stress resistance and persistence of *Salmonella* in the food industry.

## 1. Introduction

*Salmonella* can cause diarrhea, vomiting, fever, abdominal cramps, and other illnesses in humans. *Salmonella enteritidis* and *Salmonella typhimurium* are the most significant serotypes associated with human infections [1]. Despite the implementation of interventions aimed at reducing contamination during slaughter and processing, beef remains a common source of *Salmonella* outbreaks [2]. During beef cattle slaughter and processing, microorganisms found on fresh carcasses typically originate from feces, or hides [3,4]. A slightly acidic environment is very common during the slaughter of beef cattle. After slaughter, muscle glycogen in the carcass is broken down to produce lactic acid, and ATP is decomposed to release phosphate ions, lowering the carcass pH to approximately 5.4 [5]. This mildly acidic environment facilitates acid adaptation in *Salmonella* [6]. Acid-adapted *Salmonella* not only exhibits increased resistance to similar acidic conditions and induces acid-resistant responses, but may also enhance biofilm formation, posing a serious threat to the meat processing industry and food safety [7,8]. Under various stress conditions, biofilms represent a unique mode of bacterial growth, characterized by a three-dimensional structure composed of microbial cells and extracellular polymeric substances (EPS), adhering to both abiotic and biotic surfaces [9,10]. Here, the curli fimbriae synthesis mediated by the *csgDEFG* and *csgBAC* operons, as well as the cellulose production facilitated by the *bcsABZC* and *bcsEFG* operons, constitute the essential structural components of *Salmonella* biofilm formation [11,12]. The persistent formation of biofilms is a major contributor to bacterial infections and foodborne illnesses, presenting significant challenges in both clinical and food industry settings [13]. Biofilms enhance bacterial tolerance to harsh environmental conditions, making them more difficult to eliminate than their planktonic counterparts [14].

The role of the two-component system (TCS) is well recognized as a pivotal function in biofilm development and the regulation of virulence in numerous pathogenic bacteria [15,16]. In particular, the PhoP/PhoQ TCS is a key signal transduction regulator that senses environmental signals such as low Mg^2+^, acidic pH, antimicrobial peptides, and osmotic stress, thereby controlling genes involved in stress tolerance, virulence, adhesion, and invasion [17,18,19]. PhoQ, a membrane-bound histidine kinase, senses a variety of environmental signals and initiates a phosphorylation cascade that activates the response regulator PhoP, ultimately modulates the transcription of downstream target genes [20]. In general, the absence or inhibition of PhoP/PhoQ TCS function leads to reduced biofilm formation [21] and decreased antibiotic resistance in biofilm cells [22]. Previous studies have shown that the PhoP/PhoQ TCS plays diverse regulatory roles in biofilm formation across species, through the regulation of membrane proteins [23], motility [24], and polysaccharide biosynthesis [25]. However, although certain genes exist in different species, they function differently in various bacteria (i.e., their regulatory rules differ), and most PhoP-regulated genes are species- or strain-specific [26]. While most evidence points to the PhoP/PhoQ system contributing to biofilm formation in bacteria, limited (if any) information is available about how it works in *S*. *typhimurium* biofilm formation, especially when the bacteria are under acid stress. Our previous research has demonstrated that under the slightly acidic conditions (pH 5.4) encountered during beef cattle slaughter, the PhoP/PhoQ system of *S*. *typhimurium* plays a crucial role in regulating multiple stress responses, including acid, heat, osmotic stress, and antibiotic resistance [27].

To elucidate further this knowledge, i.e., to the precise genes and processes involved in the PhoP/PhoQ system concerning the issue of persistent survival in *Salmonella*, this study designed to investigate the effect of PhoP/PhoQ TCS on biofilm formation in *S*. *typhimurium* after acid induction. Initially, the complete biofilm formation process of *phoP*-deficient strains and their parental strains after acid induction was monitored over a period of 1–7 d, and the content of extracellular polymers was measured. During the biofilm maturation phase, confocal laser scanning microscopy (CLSM) and scanning electron microscopy (SEM) were employed to observe the impact of *phoP* gene deletion on biofilm structure. Finally, transcriptomic analysis was conducted to explore the complete regulatory network of *phoP* in biofilm formation in acid-adapted *S*. *typhimurium*.

## 2. Materials and Methods

### 2.1. Bacterial Strains Activation and Preparation of Adapted and Non-Adapted Strains

According to the description by Lang et al. [28], a *phoP* gene knockout strain (Δ*phoP*) was constructed using *S*. *typhimurium* ATCC 14028 as the wild-type (WT) strain. All strains were stored in LB (Luria–Bertani) broth medium (Land Bridge, Beijing, China) containing 30% (*v*/*v*) glycerol at −80 °C. The strain was continuously cultured at 37 °C in LB broth medium (Land Bridge, China) for 18 h twice to be activated. Activated strains were transferred into 50 mL of LB medium adjusted to pH 5.4 (Ultimate pH of chilled beef) with 3 M hydrochloric acid (acid stress treatment group) or unmodified LB medium (non-acid stress control group) and incubated at 37 °C for 4 h, resulting in the generation of acid-adapted and non-adapted strains, respectively. Samples of each treatment (non-adapted WT, Acid-adapted WT, non-adapted Δ*phoP* and Acid-adapted Δ*phoP*) were centrifuged (Eppendorf 5804R, Hamburg, Germany) at 10,000× *g* at 4 °C for 10 min. Finally, resuspend the resulting bacterial cells in LB broth and dilute to a cell concentration of 7 log CFU/mL for use in subsequent experiments.

Additionally, activated WT and Δ*phoP* bacterial cultures were inoculated into neutral (pH 7.2) and acidic (pH 5.4) LB broth to assess whether the *phoP* gene deletion affected the growth of *S*. *typhimurium*. The bacterial cultures were incubated at 37 °C with shaking for 24 h. During this period, bacterial suspension was sampled every 2 h, diluted, and spread on Tryptose Soya Agar (TSA) (Land Bridge, China) plates. The plates were incubated at 37 °C for 24 h, after which colony counts were performed and growth curves plotted [29].

### 2.2. Acid Resistance

Acid-adapted or non-acid-adapted bacteria (1 mL, 7 log CFU/mL) were inoculated into 9 mL LB broth at pH 3.0 (adjusted by HCl) for 2 h at 37 °C for acid challenge, respectively. Bacterial cultures before (0 h) and after (2 h) acid challenge were diluted in sterile peptone solution (0.85% *w/v*) and spread on TSA medium. TSA plates were incubated at 37 °C for 24 h, and the number of viability cells was calculated at each time point. The survival rate was the ratio (%) of the number of cells (CFU/mL) after acid shock (2 h) to the number of initial cells (0 h).

### 2.3. Biofilm Formation Assay

Biofilm formation was assessed using the crystal violet staining method outlined by Yin et al. [30]. Acid-adapted and non-adapted cells (200 μL, 6 log CFU/mL) were inoculated into 96-well plates and incubated at 25 °C for 1 to 7 d. After each incubation, the plates were washed thrice to remove planktonic bacteria, then stained with 200 μL 0.25% (*w*/*v*) crystal violet for 30 min in the dark. After staining, the plates were washed again with sterile water, and 200 μL of 95% (*v*/*v*) ethanol was added to solubilise the stain. Absorbance was measured at 570 nm using a microplate reader (SpectraMax M5, San Jose, CA, USA).

### 2.4. Biofilm Cell Metabolic Activity

The metabolic activity of biofilm cells was assessed using the Cell Counting Kit-8 (CCK-8, Solarbio, Beijing, China) following the method by Liu et al. [31]. Biofilms were cultured in 96-well plates with LB medium as described in Section 2.3 and incubated at 25 °C for 3, 4, 5, and 7 d. After the incubation period, planktonic bacteria were removed, and 100 μL of sterile phosphate-buffered saline (PBS) (Solarbio, China) along with 10 μL of CCK-8 solution, was added to each well. The plates were then incubated in the dark at 25 °C for 4 h. Absorbance was measured at 450 nm using a microplate reader.

### 2.5. Cell Motility

Swimming plates were prepared as described by Yang et al. [32], according to a composition of 10 g/L tryptone, 5 g/L NaCl and 0.3% agar. 3 μL of acid-adapted or un-acid-adapted WT and Δ*phoP* bacterial solutions were inoculated in the centre of swimming plates and incubated at 25 °C for 24 h. The diameter of the motility zone (cm) was measured by vernier calliper.

### 2.6. Extracellular Polymeric Substances

A total of 1 mL of acid-adapted or non-acid-adapted WT and Δ*phoP* was cultivated in 24-well polystyrene microtiter plates (final concentration of 6 log CFU/mL). The plates were incubated at 25 °C for 3, 4, 5 and 7 d to form biofilms. After biofilm formation, the wells were washed with PBS to remove planktonic bacteria. Biofilm samples adhering to the bottom of these wells were scraped with a sterile spatula and 1 mL PBS was added to each well to collect biofilm samples. Subsequently, biofilm samples were added to 0.4 mL of 1 mol/L NaOH and shaken in a shaker (150 r/min) at 4 °C for 3 h. Then, 0.6 mL of saline solution was added to the mixture and centrifuged at 6000× *g* for 20 min. The supernatant was filtered through the 0.45 μm filter membrane and the filtrate was collected for determination of polysaccharide and protein content. Polysaccharide content in biofilms as determined by the phenol/ sulfuric acid method [33]. The protein in the biofilm content was measured using the BCA protein assay kit (Sangon Biotech, Shanghai, China) [34].

### 2.7. CLSE & SEM Analyses

For CLSM analysis, acid-adapted or non-acid-adapted WT and Δ*phoP* were cultured in 35 mm × 10 mm cell culture dishes (Sigma, Burlington, MA, USA) containing 4 mL of LB broth, achieving a final concentration of 6 log CFU/mL. The dishes were incubated at 25 °C for 4 d. After incubation, the biofilms were washed three times with sterile PBS to remove planktonic bacteria. The biofilms were stained for 30 min in the dark using the LIVE/DEAD BacLight kit L-7012 (Thermo Fisher, Waltham, MA, USA). This kit uses Syto 9 to stain intact bacterial membranes and propidium iodide to stain bacteria with damaged membranes. CLSM imaging was conducted using the LSM 880 (Zeiss, Oberkochen, Germany) with a 63 × oil immersion objective and 488 nm argon laser. Fluorescence was captured in two emission ranges: 480~500 nm for Syto 9 and 490~635 nm for propidium iodide. Three-dimensional projections were created from z-stack data using ZEN Blue Lite 2.3 software.

For SEM analysis, polystyrene sheets (2 mm thick, 8 mm diameter) were placed horizontally in 48-well plates. LB broth with acid-adapted or non-acid-adapted strains was added to each well, achieving a final bacterial concentration of 6 log CFU/mL. After incubation at 25 °C for 4 d, the sheets were washed with sterile PBS to remove planktonic bacteria. The biofilms were fixed in 2.5% glutaraldehyde for 24 h at 4 °C and then dehydrated in a series of graded ethanol concentrations (50%, 70%, 80%, 90%, and 100%) for 10 min each. Following critical point drying with liquid CO_2_ and gold coating, the samples were examined using a SU8020 SEM (Hitachi, Tokyo, Japan).

### 2.8. Transcriptomic Analysis

#### 2.8.1. RNA Extraction

RNA extraction was performed from *S. typhimurium* with acid adaptation (pH 5.4 induced for 4 h) and non-acid adaptation (pH 7.2 LB medium) in Section 2.1. Total RNA was extracted using the RNAiso Plus Kit (Takara, Beijing, China) according to the instructions provided by the manufacturer. To minimize random errors, independent triplicates were used for RNA extraction.

#### 2.8.2. RNA-Seq Analysis

RNA quality and integrity were assessed using the RNA Nano 6000 Assay Kit and the Bioanalyzer 2100 system. Once qualified, the libraries were pooled based on effective concentration and target sequencing data. The raw sequencing data was assessed for quality using FastQC (v0.10.0). Clean reads were aligned to the reference genome obtained from the NCBI RefSeq database (Accession number: GCF_000006945.2). Genes with Reads Per Kilobase per Million mapped reads (RPKM) values greater than 20 were considered highly expressed. Differential expression analysis was conducted between three groups: Δ*phoP* vs. WT, Δ*phoP*-acid-adapted vs. Δ*phoP*, and Δ*phoP*-acid-adapted vs. WT-acid-adapted. Adjusted *p*-values (Benjamini–Hochberg method) identified significantly differentially expressed genes at|log_2_ (fold change)|> 1 and *p*-adj < 0.05. Kyoto Encyclopedia of Genes and Genomes (KEGG) and Gene Ontology (GO) enrichment analyses were performed on differentially expressed genes (DEGs).

### 2.9. Quantitative Real-Time Polymerase Chain Reaction (qRT-PCR)

Total RNA was extracted from biofilms in each group as described in Section 2.8. RNA was used to synthesize complementary DNA with the Evo M-MLV reverse transcription kit (Accurate, Hunan, Changsha, China). The mRNA levels of biofilm-associated genes were quantified using SYBR Green Pro Taq HS Premix (Accurate, Hunan, Changsha, China) on a 96-well Real Time PCR System (Bio-Rad, Hercules, CA, USA). Primer sequences for each gene (Table A1) were designed using published GenBank sequences and the Primer3 Plus. The 16S rRNA gene was used as an internal reference.

### 2.10. Statistical Analysis

Each experiment was repeated on three separate occasions with three duplicates on each occasion. Data were presented as the mean ± standard error. Differences between the WT and Δ*phoP* strains were analyzed using the MIXED procedure (SAS, Version 9.0, Cary, NC, USA). Fixed factors included strain, acid adaptation, and incubation time, while random factors consisted of experimental replicates. The least square means for type-3 tests of fixed effects were separated using the PDIFF option and were considered significant at *p* < 0.05. Gene expression levels were analyzed using the 2^−ΔΔCt^ method.

## 3. Results

### 3.1. Acid Tolerance Response

Compared to the WT, Δ*phoP* exhibited no obvious growth defects whether in acidic or non-acidic environments (Figure A1). Based on the above, this study proceeded to further acid challenge tests. Acid-adapted or non-acid-adapted WT and Δ*phoP* were acid-challenged under strong acid condition at pH 3.0, and their survival rates are shown in Figure 1. The survival rates of WT and Δ*phoP* under strong acidic condition without acid adaptation were 0.14% and 0.43%, respectively. After acid induction in a slightly acidic environment at pH 5.4, both strains developed the acid tolerance effect with significantly higher survival rates. In the control group, the survival rate was increased to 33%, while *phoP* deletion led to a significant reduction in acid tolerance to only 2.38%. This suggested that *phoP* was activated under slightly acidic condition and had a crucial role in the acid tolerance response of *S. typhimurium*.

### 3.2. Biofilm Formation Ability

The ability of *S*. *typhimurium* to form biofilms is significantly affected (*p* < 0.05) by the interaction of three factors: incubation time, acid adaptation and *phoP* gene (Table 1). Biofilm formation tended to increase significantly with increasing incubation time (*p* < 0.05). For WT, acid-adapted strains were able to form more biofilm structures in the later stages of incubation, and the biofilm-forming ability of WT in the acid-adapted state was significantly higher than that of the non-adapted strains after 4 d. The crystal staining values of non-adapted and acid-adapted WT biofilms increased from 0.15 and 0.16 on the 1st d to 2.14 and 3.10 on the 7th d, respectively. Notably, there was no significant difference in biofilm formation in *S*. *typhimurium* in the absence of acid signalling, whether the *phoP* gene was deleted. However, under acid-induced conditions, deletion of the *phoP* gene resulted in a significant reduction in the biofilm-forming ability of *S*. *typhimurium*, especially after 4 d. Until incubated for 7 d, the biofilm biomass of Δ*phoP* after acid adaptation was 67.10% of that of WT, indicating that the deletion of *phoP* gene affects the maximum amount of biofilm formed after acid adaptation. Thus, *phoP* not only has an important role in responding to acid signalling, but also in the regulation of biofilm development when it is activated. In summary, 4 d was the critical time point for biofilm formation of acid-adapted strains in this study; therefore, it will be identified as the critical time point to determine other indicators in subsequent experiments.

### 3.3. Biofilm Metabolic Activity

Biofilm metabolic activity of *S*. *typhimurium* is significantly (*p* < 0.05) affected by the interaction of three factors: incubation time, acid adaptation and *phoP* gene (Table 2). In WT, both biofilm cells of strains in the non-acid-adapted and acid-adapted states maintained high cellular metabolic activity during the biofilm growth period (4–7 d), which was higher in acid-adapted WT. After *phoP* deletion, there was no significant difference between the metabolic activities of WT and Δ*phoP* in the early stages of biofilm development (3–4 d) without acid adaptation, and there was no significant increase in the metabolic activity of biofilm cells with Δ*phoP* after 4 d. After acid adaptation, the cell metabolic activity of Δ*phoP* was lower than that of acid-adapted WT during the period of biofilm development (4–7 d), while this is also consistent with the entire trend of biofilm formation. These results suggested that the *phoP* gene is a key regulatory locus for enhancing the metabolic activity of biofilm cells under acid-induced conditions.

### 3.4. Motility

Motility of *S*. *typhimurium* was significantly affected by the acid adaptation and the *phoP* gene deletion (Figure 2). Acid-adapted treatments significantly improved the swimming ability of WT and Δ*phoP*, and the diameter of WT and Δ*phoP* movement increased by 2.03 and 0.30 cm, respectively, after acid adaptation. However, compared to acid-adapted WT, the swimming ability of Δ*phoP* was significantly reduced due to the deletion of *phoP*, and the colony diameter of Δ*phoP* on semi-solid medium was only 58.6% of that of WT. This indicated that both acid adaptation and *phoP* contributed to the swimming ability of *Salmonella*.

### 3.5. Extracellular Polymeric Substance

The interaction of inoculation time, acid adaptation status and the absence or not of the *phoP* gene on EPS in *S. typhimurium* biofilms was significant (Table 3). Changes in EPS with incubation time were consistent with biofilm formation, with all strains, whether acid-adapted or not, growing significantly (*p* < 0.05) after 4 d and beginning to form biofilm structures. Moreover, for WT, the extracellular polysaccharide and protein contents of strains in the acid-adapted state were significantly (*p* < 0.05) higher than those in the non-acid-adapted WT from 4 to 7 d. This indicated that acid adaptation promoted the production of extracellular polymers.

After deletion of the *phoP* gene, the extracellular polysaccharide content of acid-adapted Δ*phoP* was significantly (*p* < 0.05) higher than that of non-acid-adapted Δ*phoP* only at the 4th and 5th d. Its extracellular polysaccharide content on 7 d of incubation and its extracellular protein content during the entire incubation period (4 to 7 d) were not significantly different (*p* > 0.05) from that of the non-acid-adapted Δ*phoP*. Acid adaptation treatment did not enhance the production of extracellular proteins of Δ*phoP*. More importantly, the EPS production of Δ*phoP* throughout the incubation period (4–7 d) after acid adaptation was significantly (*p* < 0.05) lower than that of WT after acid adaptation. The above results indicated that *phoP* gene and acid adaptation treatment promote biofilm formation of extracellular polymeric substances in *S*. *typhimurium*.

### 3.6. CLSM & SEM

To explore the effect of *phoP* gene deletion and acid adaptation on the microstructure of *S*. *typhimurium* biofilm, the distribution of biofilm cells 4 d was observed by SEM (Figure 3) and CLSM (Figure 4), respectively. Under SEM at 5000× magnification, without acid-adapted treatment, the distribution of bacteria was more dispersed, with only a small amount of secretion on the surface of the bacteria, and there was no aggregation among the bacteria to form agglomerated structures (Figure 3(A1,A2)). After acid adaptation, WT and Δ*phoP* bacteria secreted more EPS on their surfaces, and the bacteria aggregated with each other and grew in clusters (Figure 3(B1,B2)). However, upon *phoP* deletion, a reduction in the cluster-like biofilm structure was clearly observed, and the acid-adapted state of Δ*phoP* (Figure 3(B2)) secreted fewer EPS than the acid-adapted WT (Figure 3(B1)). This suggested that acid induction and the *phoP* gene were beneficial in promoting EPS production in *S*. *typhimurium* biofilms, which was also consistent with the results of EPS content measurements.

CLSM analysis was further performed by labelling live and dead cells with green and red fluorescence by SYTO-9 and PI probes, respectively (Figure 4). After acid adaptation, The WT strain formed a dense biofilm layer and the number of viable bacteria in the biofilm increased (Figure 4(B1)), compared to non-acid-adapted WT (Figure 4(A1)). In contrast, acid-adapted Δ*phoP* strains had significantly reduced biofilms (Figure 4(B2)), compared to acid-adapted WT (Figure 4(B1)). Acid adaptation could enhance the attachment of *S*. *typhimurium*, whereas *phoP* gene deletion reduced cells attachment.

### 3.7. RNA-Seq

#### 3.7.1. GO Enrichment Analysis

The DEGs were annotated to the GO database to obtain information about the possible functions of DEGs, which included biological process (BP), cellular component (CC) and molecular function (MF). In the absence of acid adaptation, multiple functional annotations involved in *S*. *typhimurium* PhoP/PhoQ TCS were enriched for down-regulation due to the absence of *phoP* (Figure 5A). In this study, the BP that showed enrichment in Δ*phoP* compared to the WT primarily includes intracellular signal transduction (GO:0035556), phosphorelay signal transduction system (GO:0000160), and cellular protein modification processes (GO:0006464). In terms of CC, the enriched categories included components of the outer membrane (GO:0016021), external encapsulating structures (GO:0030312), among others. Regarding MF, the enriched categories were amino acid transmembrane transport activity (GO:0015171), transporter enzyme activity, acyl group transferase activity (GO:0016746), and metal ion transmembrane transport activity (GO:0046873). Thus, the deletion of *phoP* first affects membrane components, including *pagC*, *pagO*, *pagP*, *basS*, *araE*, *ybfM*, *kgtP*, etc. Meanwhile, *phoP*, as a response regulator of the TCS, gene deletion reduced *S*. *typhimurium* signalling (*rstA*, *phoQ*, *phoP*, *prpR*, *arcA*, etc.) and transmembrane transporter protein activities (*potE*, *ycaM*, *brnQ*, *yjeH*, *yaaJ*, etc.) in BP and MF, respectively. Deletion of *phoP* may lead to alterations in signal transduction and transmembrane transport processes, which are essential for biofilm formation, by affecting key genes involved in membrane integrity and transport.

Compared to the non-acid-adapted Δ*phoP*, the acid-adapted Δ*phoP* showed significant enrichment in BP for peptide biosynthetic process (GO:0043043), locomotion (GO:0040011), and response to external stimulus (GO:0009605), among others. In CC, significant enrichment was found in organelle (GO:0043226) and cytoplasmic part (GO:0044444). In MF, significant enrichment was observed in structural molecule activity (GO:0005198), RNA binding (GO:0003723), and ion binding (GO:0043167), among others (Figure 5B). These activities reflect the adaptive mechanisms of acid stress response, where the acid-adapted Δ*phoP* enhances peptide synthesis, motility, and external stimulus response, alongside strengthening cellular structures and ion regulation, to better cope with the environmental stress.

However, Δ*phoP* showed limited mobilization in multiple genes during acid adaptation compared to the acid-adapted WT (Figure 5C). These down-regulated DEGs were significantly enriched in BP to oxidation-reduction process (GO:0055114), response to stimulus (GO:0050896), etc.; in CC to external encapsulating structure (GO:0030312), periplasmic space (GO:0042597), etc.; and MF was enriched for cofactor binding (GO:0048037). After acid adaptation, the redox and stress response ability of Δ*phoP* was significantly reduced compared with that of WT, which may further reduce the stress resistance and biofilm formation ability.

#### 3.7.2. KEGG Enrichment Analysis

The regulatory pathways involved in DEGs were further obtained by pathway significant enrichment, and the results are shown in Figure 6. In the absence of acid adaptation, the ∆*phoP* mutant relative to the WT was enriched with 30 (*ybjZ, cysW*, *mglB*, etc.) and 21 (*yfbE*, *rstA*, *phoQ*, *phoP*, *pagC*, etc.) DEGs in the ABC transporters (seo02010) and two-component system (seo02020) pathways, respectively, which are related to environmental information processing. Furthermore, the ∆*phoP* mutant was enriched with 6 (*ybjG*, *uppS*, *dacC*, *dacB*, etc.) and 15 (*lpxO*, *rfaQ*, *yrbH*, *yjdB*, *pagP*, etc.) DEGs in the peptidoglycan biosynthesis (seo00550) and lipopolysaccharide (LPS) biosynthesis (seo00540) pathways, which are involved in membrane modification. Additionally, 16 DEGs (*pmrD*, *yfbE*, *pmrF*, *yfbG*, *phoQ*, *phoP*, etc.) were enriched in the cationic antimicrobial peptide (CAMP) resistance (seo01503) pathway (Figure 6A). This suggests that *phoP* deletion reduces mechanisms related to environmental sensing, membrane biosynthesis and stress resistance in *S*. *typhimurium*.

Under acid-inducing condition, compared to the non-acid-induced ∆*phoP*, the acid-adapted ∆*phoP* showed upregulated pathways mainly enriched in those related to peptide biosynthesis and motility, such as ribosome (seo03010) (*rpmE2*, *rpsR*, *rplI*, *rpsF*, *rplA*, etc.), bacterial chemotaxis (seo02030) (*tcp*, *motA*, *motB*, *cheM*, *cheA*, etc.), and flagellar assembly (seo02040) (*flgM*, *flgN*, *flgK*, *fljB*, *flgL*, etc.), which were enriched with 38, 18, and 16 DEGs, respectively. The TCS (seo02020) pathway, associated with signal transduction, was enriched with 46 DEGs. Additionally, metabolic processes related to global and overview metabolism, such as nitrogen metabolism (seo00910), pyruvate metabolism (seo00620), and carbon metabolism (seo01200), were also identified (Figure 6B). These pathways represent the mechanisms and compensatory responses of ∆*phoP* mutants to acid stress following *phoP* deletion, highlighting changes in peptide biosynthesis, motility, signalling and metabolism.

Under the same acid-adapted conditions, compared to WT, the deletion of *phoP* led to a significant downregulation of the CAMP resistance (seo01503) pathway (*yfbE*, *pmrF*, *yjdB*, *basR*, *phoP*, *pqaB*, *phoQ*, etc.). Furthermore, various global and overarching metabolic pathways, such as the citrate cycle (TCA cycle) (seo00020), carbon metabolism (seo01200), and microbial metabolism in diverse environments, were enriched in downregulated genes. Pathways related to amino acid metabolism, including lysine degradation (seo00310), cysteine and methionine metabolism (seo00270), and valine, leucine, and isoleucine degradation (seo00280), were also significantly downregulated. Moreover, pathways related to cell motility, such as bacterial chemotaxis (seo02030), and environmental information processing, such as ABC transporters (seo02010), were also enriched in downregulated genes, with 54 and 11 DEGs, respectively (Figure 6C). These pathways highlight the limited mobilization of the process after *phoP* deletion compared to WT, suggesting that although some of the pathways in ∆*phoP* were up-regulated under acid stress, those do not show the same level of activation as WT, which shows to a partial attenuation of the stress response.

### 3.8. qRT-PCR

In this study, 10 *S*. *typhimurium* stress resistance-related genes were selected for qRT-PCR analysis based on RNA-seq and further validated the results of RNA-seq (Figure 7). The relative expression of genes in the non-acid-adapted ∆*phoP* mutant strain, the acid-adapted WT and the acid-adapted ∆*phoP* mutant strain were determined using the gene expression of the non-acid-adapted WT as a standard. The deletion of the *phoP* gene in *S*. *typhimurium* had an effect on the expression levels of flagellar and biofilm master regulator (*flhC*, *csgD*), virulence-related genes (*invA*, *hilA*, *hilD*), TCS-related genes (*ssrB*, *ompR*) and stress response-related gene (*iraM*, *rpoS*, *luxS*) in the Δ*phoP* were lower than those in the WT strain (*p* < 0.05). After acid induction, the expression of the measured genes was upregulated in both WT and Δ*phoP* mutant strains. Still, the gene expression of Δ*phoP* was significantly lower than that of WT in each case due to the absence of *phoP*. The qRT-PCR results confirmed that the deletion of *phoP* in *S*. *typhimurium* reduced the expression of key stress resistance-related genes, with trends consistent with the RNA-seq data.

## 4. Discussion

The PhoP/PhoQ TCS is ubiquitous in Gram-negative bacteria and is associated with stress response and biofilm formation [23,24,25]. In this study, the Δ*phoP* mutant of *S*. *typhimurium* was constructed for analysis. The sensor histidine kinase PhoQ is sensed at acidic pH and mediates PhoP-dependent gene activation [18,35]. The results of this study showed that the acid tolerance response of *S*. *typhimurium* could be induced at the ultimate pH of chilled beef (pH 5.4), and the survival rate of acid-adapted strains in the acid-challenged environment was enhanced (Figure 1). Moreover, the biofilm-forming ability of *S*. *typhimurium* was significantly enhanced after acid adaptation (Table 1), while the acid tolerance and biofilm-forming ability of the Δ*phoP* mutant strain were decreased. This indicated that the absence of the *phoP* gene reduced the ability of *S*. *typhimurium* to respond to acid signalling in the stress environment via PhoP/PhoQ TCS and further affected its biofilm formation. This was also confirmed by the decrease in the swimming ability of Δ*phoP* (Figure 2) and the decrease in the ability to produce EPS (Table 3) under the same acid-induced conditions, compared to WT. It is well known that *Salmonella* flagellar motility mediates bacterial chemotaxis, which gives the bacteria an environmental advantage and is responsible for the search for suitable attachment sites during initial adhesion to biofilms [36,37]. Subsequent EPS production serves as a physical barrier that enhances the resistance of *Salmonella* to external stresses and promotes initial adhesion and stabilization of the biofilm [38,39]. Extracellular polysaccharides and proteins constitute the principal components of EPS [40]. Therefore, in the microstructural observation of acid-induced Δ*phoP* biofilm, it did not form larger clustered and aggregated cellular structures as compared to WT, which was replaced by a more dispersed biofilm (Figure 3). This is consistent with the findings of Ma et al. [24], who constructed the *Cronobacter sakazakii* Δ*phoPQ* deletion mutant for biofilm observation. Compared with WT, the Δ*phoPQ* strain exhibited reduced ATR-FTIR spectral peaks corresponding to extracellular proteins (1544 and 1649 cm^−1^) and polysaccharides (1080 and 1056 cm^−1^), and SEM revealed a sparse three-dimensional biofilm structure.

Notably, acid induction increased the metabolic activity of *Salmonella* biofilm cells, while *phoP* gene deletion resulted in a significant reduction in the metabolic activity of cells within the biofilm (Table 2). Similarly, in a study by Ma et al. [24], it was found that the absence of PhoP/PhoQ TCS function significantly reduced the metabolic activity of *C*. *sakazakii* biofilm cells. The formation of biofilms requires cell surface modification, adhesion, and EPS production, all of which are extremely energy-consuming processes [39]. Based on this result, it is likely that deletion of the *phoP* gene impairs the ability to mobilize sufficient energy, thereby limiting *Salmonella* motility and production of EPS, which are essential for biofilm formation. Although Δ*phoP* showed almost no growth defects under favourable conditions, it exhibited growth retardation and decreased survival in an acidic environment compared to WT strains, which indicates that Δ*phoP* is defective in the survival pathways it can mobilize in response to stressful environments [41].

Therefore, to investigate the potential contribution of PhoP/PhoQ TCS to biofilm formation in *S*. *typhimurium,* RNA-seq was performed in this study to identify the DEGs and their differential pathways in WT and Δ*phoP* strains before and after acidic induction. It was found that in the absence of acid adaptation (Figure 6A), *phoP* deletion resulted in significant downregulation of enriched gene functions mainly for signal transduction (GO:0007165), e.g., *rstA*, *phoQ*, *phoB*, *basS*, etc., and membrane part (GO:0044425), e.g., *ybfM*, *pagC*, *yjdB*, *pagO*, *pagP*. In KEGG analyses, these genes were mainly involved in the envelope modification-related LPS biosynthesis (seo00540) and peptidoglycan biosynthesis (seo00550) pathways, as well as the signalling-response-related two-component system (seo02020) pathway. Among them, the downregulation of *pmrD*, *phoQ*, *phoP*, *yjdB*, *pagP*, *basS* genes further led to downregulation of the CAMP resistance pathway (seo01503). This result is consistent with the regulatory function of PhoP/PhoQ TCS unearthed in previous studies. For *Salmonella*, the PhoP/PhoQ TCS is an important system required to regulate intracellular survival and resistance to CAMP. Defective function of the PhoP/PhoQ TCS often results in reduced *Salmonella* virulence, susceptibility to a wide range of antimicrobial peptides, as well as a reduced ability to respond to environmental signals and induce modifications in lipid A [42,43,44]. For example, mass spectrometry studies by Guo et al. [43] showed that *S*. *typhimurium* PhoP/PhoQ TCS modulates the structural modification of lipid A in the LPS host signalling fraction through the addition of aminoarabinose and 2-hydroxycrotonate, and that this modification alters LPS-mediated endothelial cell adhesion as well as the modulation of immune responses. Gunn et al. [42] further demonstrated that PhoP-PhoQ is a global regulator of the production of multiple envelopes or secreted proteins by *S*. *typhimurium* by studying the PhoP/PhoQ TCS-regulated *pag* loci (*pagA* and *pagE*-*P*).

However, PhoP/PhoQ TCS seemed not to be activated under normal culture conditions, and Δ*phoP* mutant strain did not show significant differences in acid tolerance (Figure 1), swimming ability (Figure 2), and biofilm formation (Table 1) compared to WT. In contrast, after acid adaptation in a slightly acidic stress environment, the Δ*phoP* mutant strain had significant defects in biofilm formation ability, suggesting that PhoP/PhoQ TCS has an important role in biofilm formation in low pH environments. After acid adaptation (Figure 6C), Δ*phoP* mutants were enriched for down-regulated pathways compared to WT, and CAMP resistance (*yfbE*, *pmrD*, *yjdB*, *basR*, *basS*, *phoP*, *phoQ*, *pagP*, etc.) was still the most significantly enriched pathway, with energy metabolism-related pathways also being the theme of the down-regulated pathway. These pathways include basic metabolic pathways such as the TCA cycle (seo00020) and pentose phosphate pathway (seo00030), multiple amino acid metabolism (seo00270, seo00260) and degradation (seo00310, seo00280), and the bacterial chemotaxis (seo02030), among others.

When bacteria are in an acidic environment, the integrity of the cell membrane and proton motive force are disrupted, leading to an excess of protons inside the cell and inhibiting bacterial growth [45]. To avoid cell membrane damage caused by acidic environments, *Salmonella* upregulates genes encoding membrane proteins as an important strategy to enhance acid resistance [46].

In this study, the deletion of *phoP* led to a significant downregulation of genes involved in LPS biosynthesis, both before and after acid induction, likely impairing biofilm formation. Notably, the genes *pagP*, which encodes lipid A palmitoyltransferase, and *yjdB*, encoding lipid A phosphoethanolamine (pEtN) transferase, were significantly downregulated. The structure of LPS is crucial for bacterial adhesion and biofilm formation [47]. Acylation modifications to LPS contribute to the hydrophobicity of the bacterial surface, enhancing its adaptability and growth in different environments [48]. In this study, the downregulation of *pagP* and *yjdB* suggests defects in LPS modification, which may have led to the observed reduction in biofilm formation.

In addition, the signal transduction protein PmrD, which is directly regulated by PhoP, was also downregulated following the deletion of *phoP*. PmrD plays a pivotal role in activating the PmrA/B TCS [49], which regulates LPS modification under acidic conditions. This process mainly involves the addition of 4-amino-4-deoxy-1-arabinose and pEtN to the phosphate group of lipid A [50,51,52]. Correspondingly, genes involved in this modification, such as *yjdB*, *yfbG*, and *yfbE* [53], were downregulated. Moreover, the BasSR TCS, a global transcription factor implicated in biofilm regulation [54], was also downregulated in the Δ*phoP* mutant. These findings suggest that the impaired sensing and regulatory systems in *Salmonella*, coupled with the reduced ability to modify the cell envelope, resulted in a biofilm lacking the necessary surface hydrophobicity and structural integrity. Consequently, this diminished *Salmonella*’s capacity to adapt to environmental changes and maintain cellular function, hindering the normal formation of biofilms.

On the other hand, the absence of *phoP* may lead to disruption of signal transduction pathways, resulting in impaired energy metabolism and bacterial chemotaxis. Compared to the WT strain, the Δ*phoP* mutant showed impaired stress response regulation in acidic environments, leading to reduced flagellar function and metabolic activity. Lang et al. [28] demonstrated that the PhoP/PhoQ TCS promotes acid tolerance of *Salmonella* in the pH environment of fresh chilled beef by regulating the amino acid metabolism system. In the present study, the Δ*phoP* strain, under acid induction, exhibited downregulation of several amino acid metabolic pathways, including arginine and lysine metabolism. This downregulation extended to broader metabolic networks, such as carbon metabolism (seo01200) and microbial metabolism in diverse environments (seo01120), which are critical for bacterial motility and energy production [55]. Meanwhile, energy metabolism pathways are also important factors supporting the initial adhesion of biofilm cells [32]. The RNA-seq analysis revealed that genes associated with bacterial chemotaxis, including *cheY*, *cheM*, *cheB*, and *cheR*, were significantly downregulated. This reduction in chemotaxis-related genes is consistent with findings by [56], who observed similar defects in *C. sakazakii* in amino acid-deficient media, impairing flagella formation and chemotaxis. Flagella, as the primary motility and attachment organ of *Salmonella*, play an essential role in the initial adhesion of biofilm cells [57]. Fan et al. [58] also demonstrated that *cheY* gene knockout strains exhibit defects in flagellar rotation, adhesion, and lower hydrophobicity, all of which are detrimental to biofilm formation. In addition to these pathways, other key genes regulating biofilm formation, such as the stress response factor RpoS [59] and the quorum-sensing (QS) molecule [60,61], were also identified as potentially influencing the bacterial capacity to adapt to external stresses and regulating biofilm formation mechanisms. For instance, Yan et al. [62] found that the PhoP/PhoQ TCS acts as a regulator of the PcoI/PcoR QS system in *Pseudomonas fluorescens* strain 2P24, thereby modulating group behaviours.

However, it should be noted that the enhanced biofilm formation observed after 4 d in this study may result from persistent physiological changes induced by acid adaptation, such as altered gene regulation and stress tolerance. Although acid exposure activates the PhoP/PhoQ TCS and promotes biofilm formation, the intervening regulatory steps require further temporal dynamic analysis to be fully unravelled.

In summary, *phoP* deficiency significantly disrupts biofilm formation in acid-induced *Salmonella*. The absence of *phoP* impairs the LPS biosynthetic pathway, resulting in a loss of the surface hydrophobicity and structural stability necessary for biofilm formation. Furthermore, the deletion of *phoP* compromises signal regulation systems, particularly under acidic conditions, leading to diminished flagellar function and metabolic activity, which in turn affects bacterial motility and adhesion. The study also suggests potential cross-regulation between multiple TCSs, highlighting the need for further research to explore whether these systems contribute synergistically to biofilm formation.

## 5. Conclusions

Preventing biofilm formation in every step of food production is critical for maintaining food safety, particularly in the food industry. This study demonstrates that in an acid-induced environment, Δ*phoP* exhibits reduced acid tolerance, motility, biofilm formation, and metabolic activity compared to acid-adapted WT strains. Under the slightly acidic conditions, RNA-seq analysis further revealed that the absence of *phoP* disrupts LPS biosynthesis, weakening surface hydrophobicity and biofilm structural integrity. On the other hand, the lack of *phoP* interferes with regulatory mechanisms in acidic environments, inhibiting (specific) metabolic pathways, flagellar function, and chemotaxis, which collectively hinder bacterial motility and adhesion. Future studies should focus on the cascade regulatory mechanisms triggered by acid-induced activation of the PhoP/PhoQ TCS to elucidate its role in *Salmonella* persistence more precisely. These findings underscore the importance of the PhoP/PhoQ TCS in regulating *Salmonella* biofilm formation and suggest that targeting this system could offer potential strategies for controlling *S*. *typhimurium* biofilms and associated food safety challenges.

## Figures and Tables

**Figure 1 foods-14-04344-f001:**
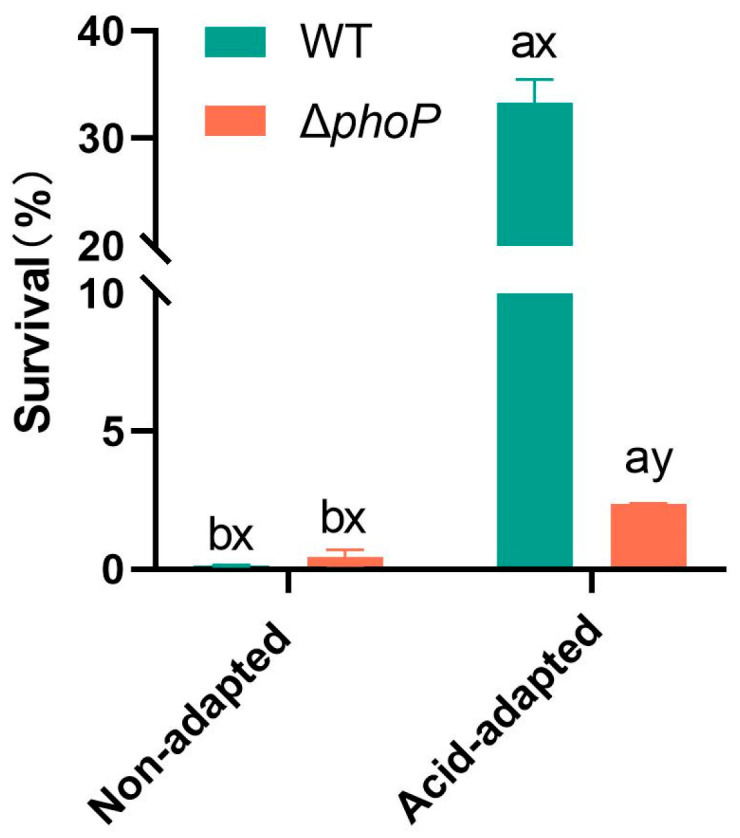
The effect of acid adaptation on the survival rate of *Salmonella typhimurium* and its *phoP*-deficient strain during acid challenge at pH 3.0. Different colours represent different experimental groups: green indicates *Salmonella typhimurium* ATCC 14028 wild-type strain (WT); red indicates *Salmonella typhimurium phoP* gene deletion strain (∆*phoP*). “a~b” indicates significant differences in survival rates of the same strain under different acid-adapted conditions. “x~y” indicates significant differences in survival rates of different strains under the same acid-adapted conditions. Data was presented as the mean ± standard error (n = 3).

**Figure 2 foods-14-04344-f002:**
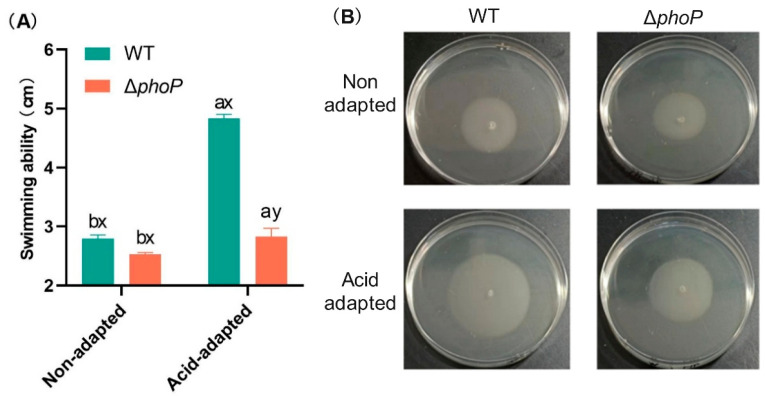
Effect of acid adaptation on the motility of *Salmonella typhimurium* and its *phoP* gene deletion strain. (**A**): Motility diameters of the wild-type strain and its *phoP*-deficient mutant on swimming plates under acid-adapted or non-acid-adapted conditions; (**B**): Colony images on swimming plates. Different colours represent different experimental groups: green indicates *Salmonella typhimurium* ATCC 14028 wild-type strain (WT); red indicates *Salmonella typhimurium phoP* gene deletion strain (∆*phoP*). “a~b”: Different letters indicate significant difference between acid-adapted and non-adapted at the same strain. “x~y”: Different letters indicate significant difference between WT and ∆*phoP* at the same stress treatment. Data was presented as the mean ± standard error (n = 3).

**Figure 3 foods-14-04344-f003:**
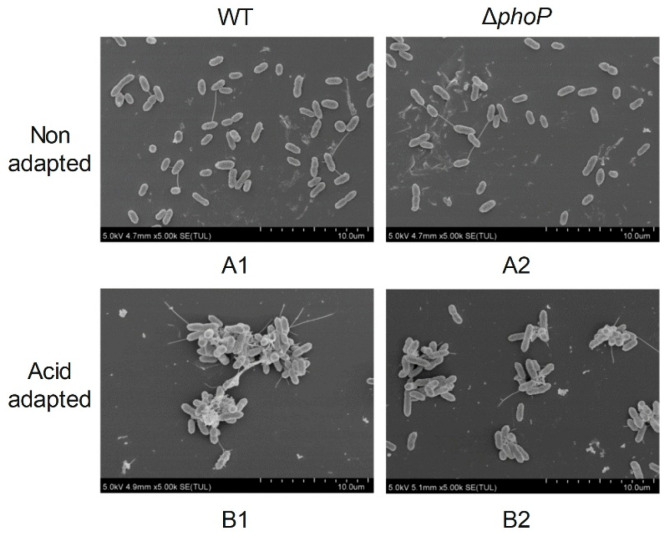
Scanning electron microscope images of different strains biofilm. WT: *Salmonella typhimurium* ATCC 14028 wild-type strain; ∆*phoP*: *Salmonella typhimurium phoP* gene deletion strain. (**A**): Non-acid-adapted strains; (**B**): acid-adapted strains. The number denotes the image arrangement sequence under acidic or non-acidic adaptation conditions.

**Figure 4 foods-14-04344-f004:**
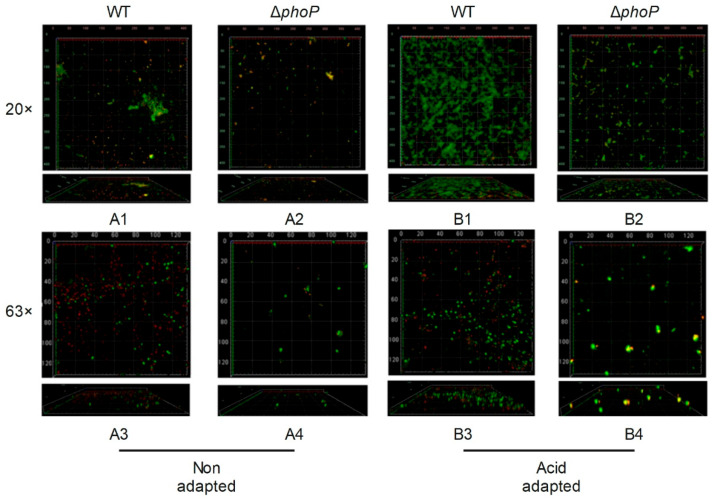
Confocal laser microscope images of different strains biofilm. WT: *Salmonella typhimurium* ATCC 14028 wild-type strain; ∆*phoP*: *Salmonella typhimurium phoP* gene deletion strain. (**A**): Non-acid-adapted strains; (**B**): acid-adapted strains; 20×, 63× indicates eyepiece magnification. The number denotes the image arrangement sequence under acidic or non-acidic adaptation conditions.

**Figure 5 foods-14-04344-f005:**
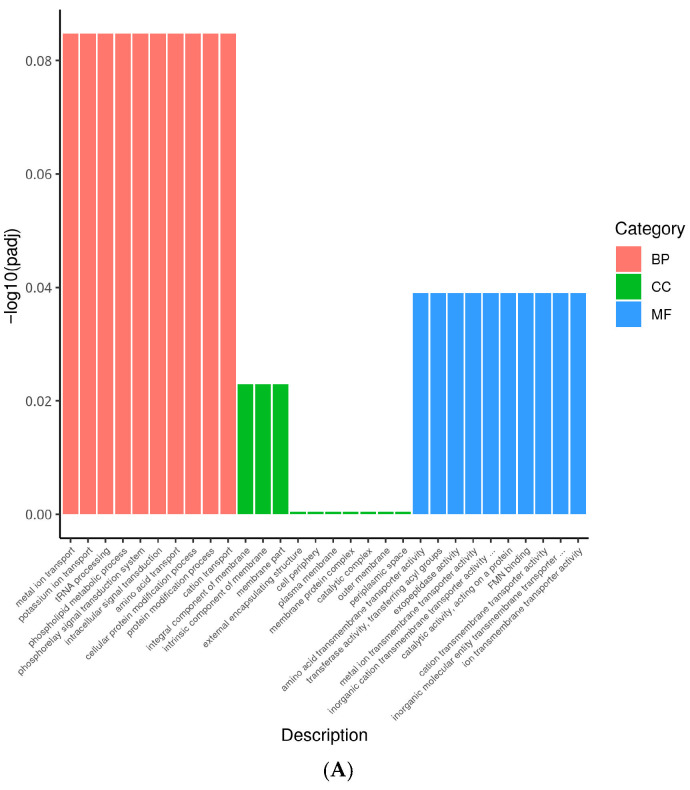
GO analysis classification of differentially expressed genes. (**A**): GO enrichment of down-regulated genes in Non-adapted ∆*phoP* vs. Non-adapted WT; (**B**): GO enrichment of up-regulated genes in acid-adapted ∆*phoP* vs. Non-adapted ∆*phoP*; (**C**): GO enrichment of down-regulated genes in acid-adapted ∆*phoP* vs. acid-adapted WT. Different colours represent different functional annotation groups: red indicates biological processes (BP); green indicates cellular components (CC); blue indicates molecular functions (MF).

**Figure 6 foods-14-04344-f006:**
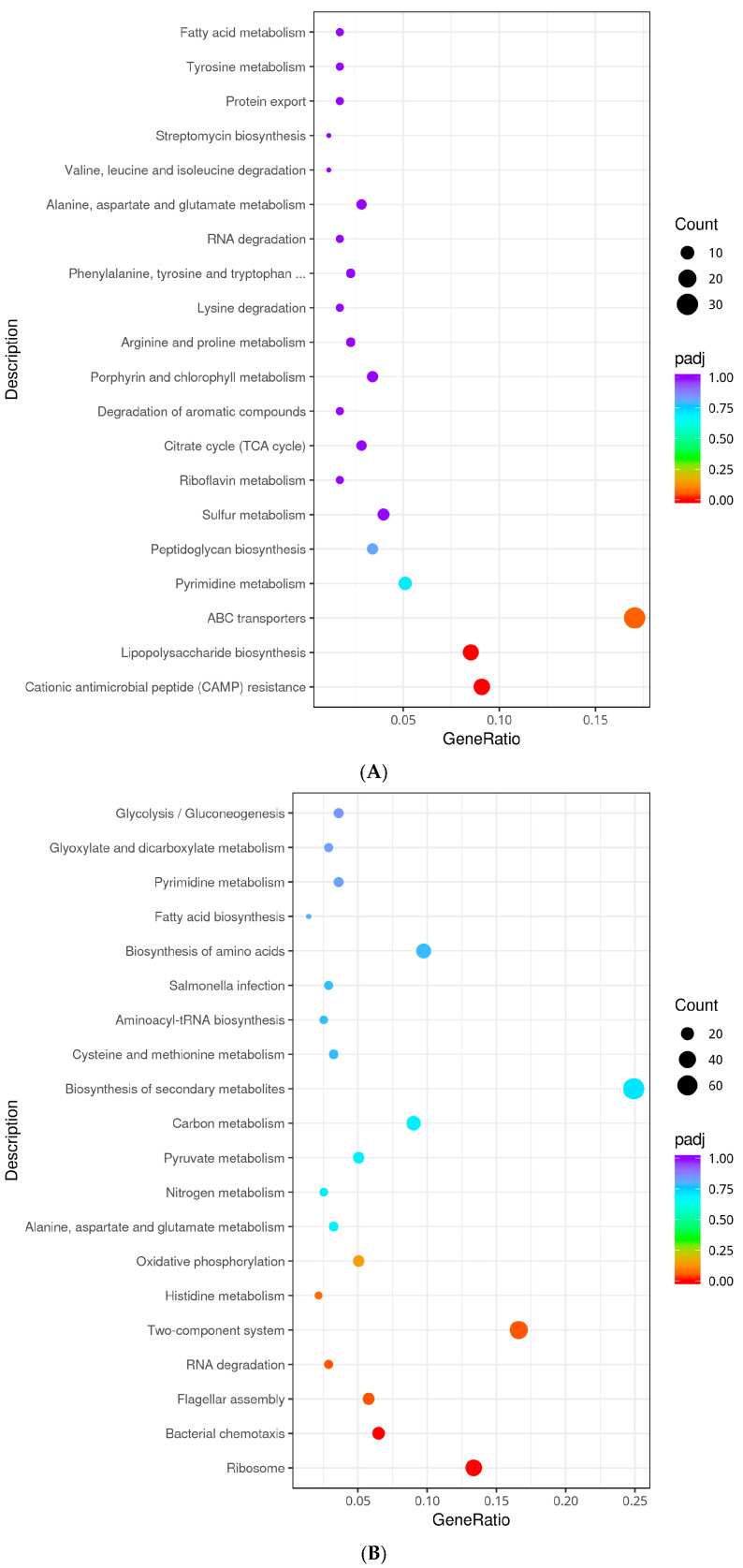
KEGG analysis classification of differentially expressed genes. (**A**): KEGG enrichment of down-regulated genes in Non-adapted ∆*phoP* vs. Non-adapted WT; (**B**): KEGG enrichment of up-regulated genes in acid-adapted ∆*phoP* vs. Non-adapted ∆*phoP*; (**C**): KEGG enrichment of down-regulated genes in acid-adapted ∆*phoP* vs. acid-adapted WT. The omitted portion in Figure A is fully titled “Phenylalanine, Tyrosine, and Tryptophan Biosynthesis.”

**Figure 7 foods-14-04344-f007:**
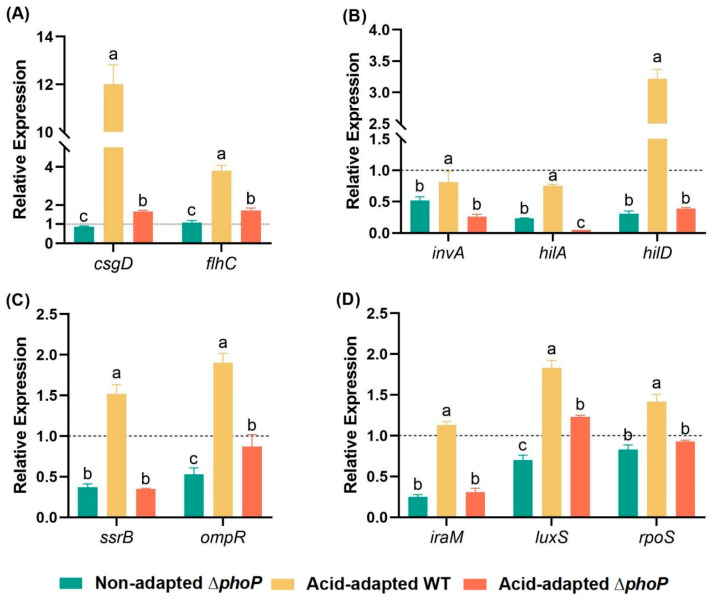
Effects of acid adaptation on flagellum and biofilm (**A**), virulence (**B**), two-component regulatory system (**C**) and stress response (**D**) gene expression in *Salmonella typhimurium* and its *phoP*-mutant strains. WT: *Salmonella typhimurium* ATCC 14028 wild-type strain; ∆*phoP*: *Salmonella typhimurium phoP* gene deletion strain. Different colours represent different experimental groups: green indicates non-acid-adapted ∆*phoP*; yellow indicates acid-adapted WT; red indicates acid-adapted ∆*phoP*. “a~c”: Different letters indicate significant difference (*p* < 0.05). Black dashed line: Non-acid-adapted WT gene expression as a baseline. Data were presented as the mean ± standard error (n = 3).

**Table 1 foods-14-04344-t001:** Effect of acid adaptation on the biofilm-forming ability of *Salmonella typhimurium* and its *phoP* gene deletion strain.

Day (d)	WT (OD_570 nm_)		∆*phoP* (OD_570 nm_)	
	Non-Adapted	Acid-Adapted	Non-Adapted	Acid-Adapted
1	0.15 ± 0.00 ^emx^	0.16 ± 0.01 ^fmx^	0.17 ± 0.00 ^emx^	0.17 ± 0.01 ^fmx^
2	0.22 ± 0.03 ^emx^	0.19 ± 0.00 ^fmx^	0.21 ± 0.02 ^demx^	0.26 ± 0.05 ^fmx^
3	0.40 ± 0.04 ^dmx^	0.42 ± 0.02 ^emx^	0.33 ± 0.01 ^cdmx^	0.43 ± 0.00 ^emx^
4	0.42 ± 0.03 ^dnx^	1.36 ± 0.04 ^dmx^	0.38 ± 0.02 ^cnx^	0.55 ± 0.00 ^dmy^
5	1.20 ± 0.05 ^cnx^	1.66 ± 0.05 ^cmx^	0.97 ± 0.03 ^bmy^	1.06 ± 0.04 ^cmy^
6	1.96 ± 0.04 ^bnx^	2.61 ± 0.08 ^bmx^	1.96 ± 0.02 ^amx^	1.87 ± 0.05 ^bmy^
7	2.14 ± 0.02 ^anx^	3.10 ± 0.10 ^amx^	2.02 ± 0.04 ^amx^	2.08 ± 0.11 ^amy^

WT: *Salmonella typhimurium* ATCC 14028 wild-type strain; ∆*phoP*: *Salmonella typhimurium phoP* gene deletion strain. a–f: Different letters indicate significant differences in the same strain and stress treatment at different times. m–n: Different letters indicate significant differences in the same strain and time at different stress treatments. x–y: Different letters indicate significant differences at the same time and stress treatment at different strains.

**Table 2 foods-14-04344-t002:** Effect of acid adaptation on the biofilm metabolic activity of *Salmonella typhimurium* and its *phoP* gene deletion strain.

Day (d)	WT (OD_450 nm_)		∆*phoP* (OD_450 nm_)	
	Non-Adapted	Acid-Adapted	Non-Adapted	Acid-Adapted
3	0.30 ± 0.02 ^bmx^	0.22 ± 0.01 ^cmx^	0.23 ± 0.07 ^bmx^	0.21 ± 0.08 ^cmx^
4	0.45 ± 0.05 ^anx^	0.97 ± 0.02 ^amx^	0.43 ± 0.03 ^anx^	0.60 ± 0.04 ^amy^
5	0.48 ± 0.01 ^anx^	0.88 ± 0.01 ^amx^	0.26 ± 0.01 ^bny^	0.38 ± 0.03 ^bmy^
7	0.45 ± 0.02 ^anx^	0.66 ± 0.06 ^bmx^	0.27 ± 0.03 ^bmy^	0.35 ± 0.02 ^bmy^

WT: *Salmonella typhimurium* ATCC 14028 wild-type strain; ∆*phoP*: *Salmonella typhimurium phoP* gene deletion strain. a–c: Different letters indicate significant differences in the same strain and stress treatment at different times. m–n: Different letters indicate significant differences in the same strain and time at different stress treatments. x–y: Different letters indicate significant differences at the same time and stress treatment at different strains.

**Table 3 foods-14-04344-t003:** Effect of acid adaptation on extracellular polymeric substances (EPS) in biofilms of *Salmonella typhimurium* and its *phoP* gene deletion strain.

EPS	Time (d)	WT		∆*phoP*	
		Non-Adapted	Acid-Adapted	Non-Adapted	Acid-Adapted
**Extracellular polysaccharides** **(ug/mL)**	4	60.64 ± 6.47 ^bnx^	141.8 ± 11.8 ^bmx^	37.32 ± 2.33 ^any^	77.00 ± 11.7 ^cmy^
	5	101.8 ± 7.32 ^anx^	206.65 ± 11.6 ^amx^	70.92 ± 1.18 ^any^	114.06 ± 3.35 ^amy^
	7	93.14 ± 7.20 ^cnx^	150.58 ± 13.2 ^bmx^	80.80 ± 8.56 ^amx^	90.28 ± 1.28 ^bmy^
Extracellular protein (ug/mL)	4	8.98 ± 1.77 ^bnx^	26.69 ± 0.94 ^cmx^	9.89 ± 4.44 ^bmx^	8.85 ± 3.43 ^bmy^
	5	12.54 ± 2.07 ^bnx^	35.97 ± 3.07 ^bmx^	8.53 ± 2.18 ^bmx^	13.36 ± 1.32 ^bmy^
	7	53.68 ± 4.12 ^anx^	63.48 ± 4.42 ^amx^	35.67 ± 4.34 ^amy^	42.15 ± 2.62 ^amy^

WT: *Salmonella typhimurium* ATCC 14028 wild-type strain; ∆*phoP*: *Salmonella typhimurium phoP* gene deletion strain. a–c: Different letters indicate significant differences in the same strain and stress treatment at different times. m–n: Different letters indicate significant differences in the same strain and time under different stress treatments. x–y: Different letters indicate significant differences at the same time and stress treatment at different strains.

## Data Availability

The original contributions presented in the study are included in the article. Further inquiries can be directed to the corresponding authors.

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
