# Peer review of "Role of the PhoP/PhoQ Two-Component Regulatory System in Biofilm Formation in Acid-Adapted Salmonella typhimurium"

_foods, 2025, doi:10.3390/foods14244344_

Round 1

Reviewer 1 Report

Comments and Suggestions for Authors

This paper attempts to research the role of phoP in acid adapted Salmonella Typhimurium biofilm formation.

There are many overlooked details in the study design, next to the poor presentation of the methods and the results. Moreover, the authors have missed some background literature.

In the introduction it is mentioned that there is no literature on PhoP and acid adaptation. Actually, a quick search shows a paper from the same group on this topic got published in May. Part of the experimental set-up was already part of this paper, as it seems.

At many other points in the intro and methods references are missing - l.43, l54, l57, l128

The methods need to be re-written. At some parts they are written as a protocol, for instance l105 - take bacterial culture.... . Also hours and h, days and d, and other inconsistencies exist. Also some media are fully described, but others are not - e.g. TSA, PSB. what are they exactly, and what brand or company. there is a repetitive sentence - l178. What is RPKM? And very important, there is no indication of replicates.

Results, also here a big problem are the replicates. there are error-bars, but what do they mean. the figure legends are incomplete, as colors used are for instance not described. Figures on sequencing are of very low quality and cannot be judged.

In the results of the biofilm formation, a sudden increase is seen in the acid adjusted situation. however, this is after 4 days, the acid adaptation cannot be the reason anymore for this increase, as the cells are already in the new situation for 4 days. nevertheless, this difference is the reasoning behind all the differences presented further in the paper between the mutant and the WT. This does not seem correct. this needs much more experimentation and explanation. 

What also is missing in this paper is a general characterization of the phoP mutant strain. How is its growth speed etc. this is very important to be able to compare it after stress conditions with a WT strain.

Moreover, the fact that the cultures are centrifuged for 10 minutes at 4 degrees before further testing is adding a complete different factor and extra stress to the comparison of the strains. The effects of this treatments should be studied separately.

Can the microscopy results somehow be quantified?

How are the biofilms actually formed? somewhere it is mentioned, cells are scraped from the bottom of the well. However, many other studies indicate air-liquid interface biofilms for Salmonella.

Comments on the Quality of English Language

The English language use needs some check-ups. It is advisable to have the paper checked by a native speaker.

Author Response

Dear Reviewer,
Thank you for your valuable comments and suggestions. We have carefully revised the manuscript accordingly. All our revisions are presented in the attached document for your review.

Best regards,
Yunge Liu

Reviewer 2 Report

Comments and Suggestions for Authors The most relevant comments are in the attached document.
However, I could mention that it is a good manuscript that needs
to improve the quality of the images, but that in general,
and with a few corrections, it could be a good publication,
due to the techniques used and the results found.

Author Response

(The authors gave the same response as above.)

Reviewer 3 Report

Comments and Suggestions for Authors

Dear Authors,

In attachment are some comments in order to improve this Manuscript.

The paper presents an overview about the role of the PhoP/PhoQ two-component system (TCS) in biofilm formation of Salmonella Typhimurium under the acid adaptation condition. After reading the submitted Manuscript, I consider that some sections can be improved. For this reason, I provide some comments that should be addressed. The following suggestions are presented:

Specific points

Line 63-73: This section would be better placed in the Discussion. It should be mentioned the mechanisms of action of the PhoP/PhoQ two-component system (TCS), the key environmental factors that influence it, and the consequences of activation.

Line 91: : Indicate the manufacturer from whom the TSB was purchased.

Line 100: Please add the full name for CFU, colony- forming units.

In general, the discussion should be improved.

Authors should discuss the results obtained by SEM analysis. Compare the obtained findings with similar assays. Highlight PhoP/Q two-component system in Salmonella as an important regulatory mechanism for resistance and its mode of action. Include future trends to keep working with the obtained data.

Author Response

(The authors gave the same response as above.)
